# Ice Particle Production in Mid-level Stratiform Mixed-phase Clouds Observed with Collocated A-Train Measurements

Damao Zhang[1], Zhien Wang[2], Pavlos Kollias[1, 3], Andrew M. Vogelmann[1], Kang Yang[2], and Tao Luo[4]

[1]Brookhaven National Laboratory, Upton, New York, USA
[2]Department of Atmospheric Science, University of Wyoming, Laramie, WY 82071, USA
[3]School of Marine and Atmospheric Sciences, Stony Brook University, New York, USA
[4]Anhui Institute of Optics and Fine Mechanics, Chinese Academy of Sciences, Hefei, Anhui, China

*Correspondence to*: Damao Zhang (dzhang@bnl.gov)

**Abstract**. Collocated A-Train CloudSat radar and CALIPSO lidar measurements between 2006 and 2010 are analyzed to study primary ice particle production characteristics in mid-level stratiform mixed-phase clouds on a global scale. For similar clouds in terms of cloud top temperature and liquid water path, Northern Hemisphere latitude bands have layer-maximum radar reflectivity (ZL) that is ~1 to 8 dBZ larger than their counterparts in the Southern Hemisphere. The systematically larger ZL under similar cloud conditions suggests larger ice number concentrations in mid-level stratiform mixed-phase clouds over the Northern Hemisphere, which is possibly related to higher background aerosol loadings. Furthermore, we show that northern mid- and high latitude springtime has ZL that is larger by up to 6 dBZ (a factor of 4 higher ice number concentration) than other seasons, which might be related to more dust events that provide effective ice nucleating particles. Our study suggests that aerosol-dependent ice number concentration parameterizations are required in climate models to improve mixed-phase cloud simulations, especially over the Northern Hemisphere.

## 1. Introduction

Ice particle production in a supercooled liquid cloud has dramatic impacts on the cloud's radiative properties, precipitation efficiency, and cloud lifetime due to distinct differences in particle sizes, shapes, fall velocities, and refractive indexes between liquid droplets and ice crystals (Sun and Shine, 1994; de Boer et al., 2011a). Such clouds significantly impact global and regional radiation budgets (Matus and L'Ecuyer, 2017) having a global coverage of more than 34% and being particularly common at high latitudes (Shupe et al., 2011; Adhikari et al., 2012; Wang 2013; Scott and Lubin, 2016). In a mixed-phase cloud, once ice particles are formed, they grow through water vapor diffusion at the expense of liquid water because saturation vapor pressure is lower over ice than liquid. This process, known as the "Wegener–Bergeron–Findeisen (WBF) process" (Wegener, 1911; Bergeron, 1935; Findeisen, 1938), creates a thermodynamically unstable condition in mixed-phase clouds. In the absence of strong vertical air motions, the WBF process removes liquid droplets quickly, causing a mixed-phase cloud to glaciate completely (Korolev and Field, 2008; Fan et al., 2011). Therefore, aerosol might impose a glaciation indirect effect on clouds by acting as effective ice nucleating particles (INPs) (Lohmann 2002). Global climate model (GCM) simulations show that changing the glaciation temperature of supercooled clouds from 0 °C to -40 °C causes differences in the top-of-atmosphere longwave and shortwave cloud radiative forcing of ~ 4 W/m$^2$ and ~ 8 W/m$^2$, respectively (Fowler and Randall, 1996).

However, observations indicate that supercooled liquid water in mixed-phase clouds persists for tens of hours or even days and down to temperatures of as low as ~-36 °C (Seifert et al., 2010; Zhang et al., 2010; de Boer et al., 2011b). Over the polar regions where mixed-phase clouds are commonly observed, cloud condensation nuclei (CCN) concentrations are usually low, on the order of 10 cm$^{-3}$ and sometimes less than 1 cm$^{-3}$ (Mauriten et al., 2011; Birth et al., 2012). An increase of aerosol may enhance CCN concentrations, thereby increase cloud cover and reduce cloud droplet size. This aerosol indirect effect leads to a longer mixed-phase cloud lifetime, which is opposite to the aerosol glaciation indirect effect (Lance et al., 2011). Even more complicated, coupling of local thermodynamic conditions and large-scale dynamics contributes greatly to the long persistence of mixed-phase clouds (Korolev and Isaac 2003; Morrison et al., 2012). Bühl et al., (2016) estimated ice mass flux at mixed-phase cloud base using ground-based radar measurements and show that when temperatures are above -15 °C the water depletion due to ice formation is small and the cloud layer is very stable. The WBF process in GCMs is typically too efficient, causing severe underestimations of supercooled liquid water fraction on a global scale (Cesana et al., 2015; McCoy et al., 2016). Tan et al. (2016) show that the equilibrium climate sensitivity (ECS) can be 1.3 °C higher in GCM simulations when supercooled liquid fractions (SLFs) in mixed-phase clouds are constrained by global satellite observations. Improved SLF parameterizations in GCMs requires better understanding of ice production processes in supercooled clouds under various dynamic environments and background aerosol conditions using extensive observations from cutting edge instruments.

Heterogeneous nucleation, which dominates ice formation in supercooled clouds at temperatures warmer than -36 °C (Pruppacher and Klett, 1997; Vali, 1996), is not well understood and parameterized in models because of the complicated

three-phase interactions of water and the largely unknown properties of ice nucleating particles (Cantrell and Heymsfield, 2005; DeMott et al., 2011; Morrison et al., 2012). There are four well-recognized heterogeneous ice nucleation models: deposition nucleation, condensation freezing, immersion freezing, and contact freezing (Pruppacher and Klett, 1997). The immersion freezing mode, which refers to the process that an INP is immersed into a droplet at a relatively warm temperature and freeze the droplet at a colder temperature, is suggested to be the dominant ice formation mechanism in stratiform mixed-phase clouds (de Boer et al. 2010). This mode provides a pathway for time-dependent ice production in clouds, which can be used to explain the long persistence of precipitating stratiform mixed-phase clouds (Westbrook and Illingworth, 2013). Of course, ice production in clouds also depends on the presence of INP and, for example, laboratory measurements of INP properties provide fundamental databases for developing and improving ice nucleation parameterizations in models (DeMott et al., 2011; Hoose and Möhler 2012; Murray et al., 2012). While such databases are valuable, it is also important to observe ice nucleation processes in the real atmosphere to constrain and evaluate parameterizations on a global scale.

Observations of aerosol impacts on ice production in supercooled clouds mainly come from ground-based and satellite remote sensing measurements. Choi et al. (2010) and Tan et al. (2014) show that supercooled liquid cloud fraction is negatively correlated with aerosol occurrence (especially dust) using Cloud-Aerosol Lidar and Infrared Pathfinder Satellite Observation (CALIPSO) spaceborne lidar measurements. Unfortunately, because the lidar signal cannot penetrate the liquid-dominated layer at the top of mixed-phase clouds, aerosol impacts on ice production are not directly presented in their studies. Seifert et al. (2010) avoided this issue by using 11 years of grounded-based lidar depolarization measurements to study relationships between dust occurrence and ice-containing cloud fractions over central Europe. Also, Zhang et al. (2012) quantitatively estimated dust impacts on ice production in mixed-phase clouds using combined CALIPSO lidar and CloudSat radar measurements over the 'dust belt', a region including the North Africa, the Arabian Peninsula and East Asia.

Our objective in this paper is to better characterize the primary heterogeneous ice production in clouds on a global scale. We focus on mid-level stratiform mixed-phase clouds, which provide a relatively simple target for studying ice generation for the following reasons. Mid-level supercooled clouds are decoupled from Earth's surface and therefore are not affected by strong turbulent vertical mixing within the boundary layer. There is usually a liquid-dominated layer at the top of mid-level stratiform mixed-phase clouds (de Boer et al., 2011b; Riihimaki et al., 2012) and, when the temperature is low enough, ice particles form from liquid droplets, grow in the water-saturated environment, and fall out of the liquid-dominated layer (Fleishauer et al., 2002; Carrey et al., 2008; Zhang et al., 2014). Below the liquid-dominated cloud layer, ice crystals continue to grow during the fall until they reach a level that is sub-saturated with respect to ice. The less complex dynamic environment and straightforward ice growth trajectory in mid-level stratiform mixed-phase clouds provides an ideal scenario for studying cloud thermodynamic phase partitioning and aerosol impacts on ice formation in clouds, as well as retrieving cloud microphysical properties with remote sensing measurements (Wang et al. 2004; Larson et al., 2006; Heymsfield et al., 2011; Zhang et al., 2012; 2014; Bühl et al., 2016). Zhang et al. (2010) present for the first time the climatology of mid-level stratiform clouds and their macrophysical properties using A-Train satellite remote sensing measurements. This study further

uses four years of collocated CloudSat radar and CALIPSO lidar measurements together with other ancillary A-Train products between June 2006 and June 2010 to provide a global statistical analysis of ice production in mid-level stratiform mixed-phase clouds.

## 2. Dataset and Methodology

The description of the collocated A-Train measurements follows directly from Zhang et al. (2010). The main instrument on CloudSat satellite is a nadir-viewing 94 GHz **C**loud **P**rofiling **R**adar (CPR)-the first spaceborne cloud radar. The sensitivity of CPR was approximately -30 dBZ during the period analyzed here. The CPR has an effective vertical resolution of about 480 m (oversampled at 240 m vertical resolution) and horizontal resolutions of between 1.3 and 1.4 km cross-track and between 1.7 and 1.8 km along-track (depending on latitude) (Stephens et al., 2008). The CPR can detect clouds with large cloud droplets, or large ice crystals, or precipitating hydrometers, and provides the vertical structures of clouds (Stephens et al., 2002). The **C**loud–**A**erosol **LI**dar with **O**rthogonal **P**olarization (CALIOP) onboard the CALIPSO satellite is a near-nadir-viewing lidar with two wavelengths at 532 nm and 1064 nm with linear polarization measurements available at 532 nm (Winker et al., 2007). The CALIOP has vertical resolutions of 30 m below 8.2 km, and 60 m between 8.2 and 20.2 km. The horizontal resolutions of CALIOP are 333 m below 8.2 km, and 1000 m between 8.2 and 20.2 km. CALIOP is able to provide global, high-resolution vertical profiles of aerosols and optically thin clouds (Winker et al., 2010). Due to differing wavelengths, the CPR and CALIOP measurements provide complementary capabilities that enable accurate detection of cloud boundaries and their vertical structures (Stephens et al., 2008). Their complementary nature is exemplified in the detection of supercooled liquid-dominated mid-level mixed-phase cloud layers, where the CPR is more sensitive to the large-sized ice crystals and the CALIOP is more sensitive to the higher number concentration of small liquid droplets (Zhang et al., 2010). Because temperature is critical for ice formation in supercooled clouds, the European Center for Medium-Range Weather Forecast (ECMWF)-AUX product is collocated to provide temperature and pressure profiles with the same vertical resolution as CPR (Partain, 2007). In addition, MODIS on the Aqua satellite provides cloud liquid water path (LWP) determined from retrieved cloud droplet effect radius and cloud optical depth (Platnick et al., 2003). The ancillary CloudSat MODIS-AUX product that includes cloud LWP is collocated and employed in our analysis. This analysis is limited to daytime hours since MODIS cloud property retrievals are only available when sunlit. Previous studies show that MODIS-retrieved LWP have a positive bias at high latitudes due to the solar zenith angle dependence in the retrieval algorithms (O'Dell et al., 2008). Through a comparison of MODIS retrievals with ground-based microwave radiometer (MWR) measurements at the Atmospheric Radiation Measurement (ARM) Facility's North Slope of Alaska (NSA) site, Adhikari and Wang (2013) show that MODIS overestimates LWP for stratiform mixed-phase clouds by 35% and 68% in the temperature ranges of -5 to -10 °C and -10 to -20 °C respectively.

Algorithms using collocated CALIOP and CPR measurements to identify mid-level stratiform mixed-phase clouds were

developed by Zhang et al. (2010). To summarize, candidate mid-level clouds are identified when the CALIOP-detected cloud top height is above 2.5 km from the ground level and cloud top temperature is greater than -40 °C. Of these clouds, many have a liquid-dominated layer at the top, which is detected by a strong peak in lidar total attenuated backscatter (TAB) near cloud top (i.e., layer maximum TAB greater than 0.06 $sr^{-1}$ $km^{-1}$) and a rapid attenuation of the lidar backscattering such that the lidar-observed layer geometric depth is less than 500 m. We use the lidar TAB and rapid lidar signal attenuation to identify the presence of liquid layers, which is a method that has been widely used for liquid layer identifications from space-borne lidar measurements (e.g., Hogan et al., 2004; Zhang et al., 2010; Wang 2013). We note that horizontally oriented ice crystals can also have a large lidar TAB however they do not attenuate lidar backscattering significantly. Wang (2013, Figure 10) shows that this approach correctly determines liquid clouds in terms of layer mean depolarization ratio and integrated backscattering coefficient. In addition, collocated MODIS cloud LWP greater than 10 $g/m^2$ is used to guarantee the detection of a liquid-dominated layer. The cloud system is identified as being stratiform when the cloud top height standard deviation is smaller than 300 m. To calculate the standard deviation, a cloud system is identified as containing at least 10 continuous cloudy profiles, which corresponds to a horizontal scale of approximately 11 km (the horizontal distance between two contiguous CPR profiles is 1.1 km). In addition, the CPR radar reflectivity factor $Z_e$ must be smaller than 10 dBZ near the surface to exclude strongly precipitating mid-level stratiform clouds.

Radar measurements are used to detect the presence of ice particles in mid-level stratiform mixed-phase clouds. Cloud droplets and pristine ice crystals are much smaller than the radar wavelength so they fall within the Rayleigh scattering regime where $Z_e$ is proportional to the sixth power of the particle size. Ice crystals are typically larger than cloud droplets such that $Z_e$ is dominated by ice crystal scattering (Shupe et al., 2007). Bühl et al. (2016) use the $Z_e$ value closest to the liquid layer base with ground-based high vertical resolution (30 m) radar measurements for studying ice particle properties. However, this is difficult with A-Train satellite measurements as the CPR has a coarse vertical resolution and the liquid layer at the top quickly attenuates CALIOP signals, preventing reliable detection of the liquid layer base. Given that the physical thickness of supercooled liquid layers at the top of mid-level stratiform mixed-phase clouds are generally smaller than 500 m and the vertical resolution of the CPR is oversampled to 240 m from the effective vertical range resolution of 480 m, we use the maximum $Z_e$ (referred to as "ZL") within 500 m below the CALIOP-detected liquid-dominated layer top to ascertain the presence of ice particles for analysis. Using temperature-dependent ZL thresholds, Zhang et al. (2010) show that, at temperatures lower than -6 °C, approximately 83.3% of mid-level liquid-topped stratiform clouds are mixed-phased, revealing the importance of understanding their ice production. Furthermore, to exclude seeding from upper-level clouds and to enable use of MODIS column integrated LWP retrievals, only single-layer clouds detected with collocated CALIOP and CPR measurements are analyzed (Wang et al., 2012). Since we study ice production in stratiform clouds in this study, we focus on clouds with top temperatures within the -40 °C to 0 °C range.

To illustrate the importance of understanding ice production in these clouds, Fig. 1 shows the global distribution of single-layer mid-level stratiform cloud occurrence during daytime based on four years of collocated CALIOP and CPR

measurements. The occurrences are smaller than what are presented in the Fig. 3 in Zhang et al. (2010) because we only focus on single-layer supercooled stratiform clouds here; while they include both single-layer and multiple-layer clouds with top temperatures warmer than -40 °C. Single-layer mid-level supercooled stratiform clouds have an annual global mean occurrence of approximately 3.3% with occurrences greater than 6% over northeastern China and the northern polar regions, and greater that 10% over the southern polar regions.

## 3. Results and Discussions

The straightforward ice crystal growth pattern in mid-level stratiform mixed-phase clouds as described above enables using $Z_e$ magnitudes to quantitatively infer ice number concentration variation in stratiform mixed-phase clouds. It is noted that because $Z_e$ is proportional to ice number concentration and also the sixth power of particle size, differences in $Z_e$ can be either attributed to large changes in ice number concentration or small changes in ice crystal size. Based on integrated *in situ* measurements and airborne remote sensing, Zhang et al. (2012) suggest that—for similar clouds in terms of cloud top temperature (CTT) and LWP—ice crystal growth in mid-level stratiform mixed-phase clouds is similar and that $Z_e$ differences reveal differences in ice number concentration. They compare ZL differences between dusty and non-dusty mid-level stratiform mixed-phase clouds and conclude that mineral dust statistically enhances ice number concentration by a factor of 2 to 6, depending on CTT. It is quite challenging to retrieve ice number concentration from radar measurements. Zhang et al. (2014) developed a method to estimate ice number concentration in stratiform mixed-phase clouds by using combined $Z_e$ measurements and 1-D ice-growth model simulations. CTT, LWP, and vertical air motion are required as inputs in their algorithms and sensitivity tests show that they all have large impacts on ice number concentration estimations. Due to large uncertainties in the MODIS-derived LWP for mixed-phase clouds (Adhikari and Wang 2013), ice number concentration estimations in mixed-phase clouds using A-Train satellite measurements are not available at this stage. In order to use the ZL magnitude to infer ice number concentration variations in this study, a narrow LWP range is selected to remove the impacts of LWP variation on the measured ZL. Fig. 2 shows the probability distribution function (PDF) of LWP for single-layer mid-level stratiform clouds from MODIS retrievals. The global mean LWP for single-layer mid-level stratiform clouds is approximately 119 $g/m^2$ with a standard deviation of 101 $g/m^2$. The PDF of LWP has a peak at approximately 45 $g/m^2$ and values decrease quickly away from the peak. For our statistical analyses, a narrow LWP range is selected from the third of the cumulative distribution centered on the LWP peak which is bounded by the values of 20 $g/m^2$ and 70 $g/m^2$.

Fig. 3 shows the global, annual-average, mid-level mixed-phase stratiform cloud ice production statistics. Fig. 3a shows the cloud distributions in terms of CTT and ZL for six latitude bands (northern and southern tropical, mid-, and high-latitudes) within the LWP range between 20 $g/m^2$ and 70 $g/m^2$. Local peaks are seen in the ZL distributions at ~ -15 °C, which correspond to the fast-planar ice growth regimes, and troughs are seen at -10 °C and -20 °C, corresponding to the relatively slow isometric growth habit (Sulia and Harrington, 2011; Zhang et al., 2014). Below -20 °C, ZL increases steadily

as CTT decreases, probably because of higher ice number concentrations at lower CTTs (Zhang et al., 2014). At a given CTT, the ZL distribution has approximately 10 dBZ variations, which might be related to different environmental aerosol loadings and/or cloud LWPs associated with each individual cloud. Nevertheless, comparing different latitude bands, the northern latitude bands statistically have larger ZL than their southern counterparts at a given CTT. The northern mid- and high latitudes have the largest ZL values.

A complementary way to view the latitudinal dependence of the cloud properties is given in Fig. 3b, which presents the mean ZL of mid-level stratiform mixed-phase clouds as a function of CTT for the narrow LWP range. Due to potential drizzle contributions to $Z_e$ measurements at relatively warm CTTs, the mean ZL is only calculated for clouds with CTT lower than -10 °C (Rasmussen et al., 2002; Zhang et al., 2017). Using mean ZL differences, we can quantitatively estimate ice concentration variations in mid-level stratiform clouds under similar cloud conditions in terms of CTT and LWP, similar to that presented in Zhang et al., (2012). From Fig. 3b, the northern mid- and high latitudes have the largest mean ZL while the southern low latitude band has the smallest values. Consistent with the cloud distribution statistics in Fig. 3a, northern hemisphere latitude bands have larger mean ZL at a given CTT than their counterparts in southern hemisphere. Depending on CTT range, the northern mid- and high latitude bands are ~ 6 and 8 dBZ larger than their southern counterparts; while the northern low latitude band is only ~ 1 dBZ greater than southern low latitude band. These results are consistent with the studies by Choi et al. (2010) and Tan et al. (2014) which show that the Northern Hemisphere has a smaller supercooled liquid fraction than the Southern Hemisphere for a given temperature range, and it is also consistent with Zhang et al. (2010) which shows that the Northern Hemisphere mixed-phase clouds have larger ice water paths (IWP).

Atmospheric pressure is another factor that could impact ice crystal diffusional growth and therefore the observed hemispheric and latitudinal ZL differences. The same subfreezing temperature at low latitudes corresponds to a higher height above mean sea level and therefore a lower atmospheric pressure level than mid- and high latitudes. Takahashi et al. (1991) show that the mass growth rate at 860 mb is approximately 30% larger than at 1010 mb due to the impact of pressure difference on the diffusivity of water vapor in air. It is also noted that, from Figure 20 in their paper, the mass growth difference due to pressure difference is much smaller than that due to temperature difference. Within the Rayleigh scattering regime, radar reflectivity is proportional to the square of ice crystal mass. Therefore, the 30% difference in mass causes approximately a 2 dBZ difference in $Z_e$. We investigated hemispheric and latitudinal differences of atmospheric pressure at subfreezing temperatures using four years of ECMWF-AUX product between 2006 and 2010. As shown in Fig. 4, for a given temperature, hemispheric differences in atmospheric pressure profiles are negligible over mid- and low latitude bands, and range from 40 mb to 140 mb over the high latitude band. Therefore, pressure-level differences have a negligible contribution to the hemispheric ZL differences over mid- and low latitude bands, and contribute less than 2 dBZ to the observed hemispheric ZL differences over the high latitude band. After removing the contributions from atmospheric pressure differences, mid-level stratiform mixed-phase clouds over northern mid- and high latitude bands still have ZL that are approximately 6 dBZ larger than their southern counterparts. By focusing on mid-level stratiform mixed-phase clouds

and carefully isolating the impacts of CTT, LWP, and atmospheric pressure, the systematically larger ZLs suggest a factor of 4 higher ice number concentrations over northern mid- and high latitudes than their southern counterparts.

The systematically larger ZL and higher ice number concentrations over the Northern Hemisphere for similar mid-level stratiform mixed clouds might be related to larger background aerosol loadings in the Northern Hemisphere. Using CALIOP measurements, Tan et al. (2104) show that the Northern Hemisphere has dramatically larger frequencies of high aerosol occurrence than the Southern Hemisphere at sub-freezing temperatures. Based on ground-based lidar and radar remote sensing measurements from sites in both Northern and Southern Hemispheres, Kanitz et al. (2011) found that layered supercooled clouds at northern mid-latitudes have significantly larger fractions of ice containing clouds compared with southern mid-latitudes, which is possibly related to the rather different aerosol conditions. In addition, larger mean ZL over the northern mid- and high latitude bands than the northern low latitude band can also be connected to larger aerosol (especially dust) loadings at sub-freezing levels. Using multiple-years of ground-based Raman lidar measurements, Seifert et al. (2010) show that Leipzig, Germany (northern mid-latitude) has much higher ice-containing cloud fraction than Cape Verde (northern low latitude) at a given CTT below 0 °C, consistent with the results in Fig.3. They proposed that possible factors influencing the differences include different sources of INP, chemical aging, as well as removal of larger aerosol particles by washout in the tropics. Indeed, although the tropics and sub-tropics have extensive dust source regions, large dust particles are cannot be elevated to sub-freezing levels without strong convection (Luo et al., 2014).

We next investigate the global impact of LWP on ZL and its latitudinal variation. At a given CTT, cloud with a larger LWP has a geometrically thicker liquid water layer, which allows ice crystals to reside longer in the liquid-dominated layer and grow larger by the WBF process. In addition, cloud with a larger LWP also has a larger ice growth rate by accretion (Zhang et al., 2014). Fig. 5 shows the mean ZL of as a function of CTT and LWP for the six latitude bands. As expected, the mean ZL increases gradually with LWP at a given CTT for all latitude bands. Mean ZL generally increases more than about 5 dBZ going from thin clouds, which are associated with small LWP, to very thick clouds, which are associated with large LWP. Therefore, observations show a dramatic impact of LWP on the measured ZL. However, within any given narrow LWP range, the mean ZL for northern latitude bands are still much greater than their southern counterparts, further supporting our conclusion that the systematic ZL differences between northern and southern latitude bands are related to aerosol activity.

To further explore aerosol impacts on ice formation, Fig. 6 shows the seasonal variations of mid-level stratiform mixed-phase cloud distributions in terms of CTT and ZL for the six latitude bands and Fig. 7 shows mean ZL seasonal variations as a function of CTT. From the statistical distributions in Fig. 6, northern latitude bands have greater ZL than their counterparts in the Southern Hemisphere at any season for similar clouds in terms of similar CTT and LWP, probably related to higher background aerosol loadings over the Northern Hemisphere. Comparing different seasons, southern latitude bands generally have little seasonal variation in ZL, as is evident in Fig. 7. In contrast, northern latitude bands have dramatic seasonal variations in ZL, with the largest ZL occurring in MAM (boreal springtime) and smallest in DJF (boreal wintertime). The

northern mid- and high latitude bands have the largest seasonal variations among all latitude bands. At CTTs warmer than -30 °C, ZLs over northern mid- and high latitude bands are larger in the boreal springtime than wintertime by approximately 4 dBZ and 6 dBZ for similar clouds in terms of CTT and LWP, respectively. From Fig. 4b, pressure profile differences between boreal springtime and wintertime are fairly small over northern mid- and high latitude bands. Therefore, the systematically larger ZLs of 4 dBZ and 6 dBZ during boreal springtime than wintertime suggests a factor of 2.5 and 4.0 higher ice number concentrations.

Dust particles are effective INPs and are recognized as one the dominant global INP sources (DeMott et al., 2010; Hoose and Möhler, 2012). Choi et al (2010) show the seasonal variation of global mineral dust occurrence at the -20 °C isotherm using the CALIOP level 2 vertical feature mask data. They observed a significant correspondence between mineral dust occurrence and reduction in supercooled cloud fraction, especially over the northern mid-latitudes, suggesting that elevated mineral dust particles effectively glaciate supercooled clouds by providing abundant INPs. In their study, the Arctic regions have dramatic seasonal variations in supercooled cloud fractions, with the lowest during the springtime. However, no obvious dust activity over the Arctic regions is shown in their paper. Luo et al. (2014) point out that CALIOP level 2 data product often misses the detection of elevated thin dust layers. They presented improved algorithms to identify thin dust layers using CALIOP layer-mean particulate depolarization ratios and CPR measurements. Fig. 8 shows the distributions of global dust occurrence and their seasonal variations at different sub-freezing temperature ranges based on the dust dataset developed by Luo et al. (2014). It is obvious that during March-April-May (MAM), the boreal springtime, northern mid- and high- latitude regions have significantly higher dust occurrences than other seasons at any given sub-freezing temperature range. Similarly, using multiple-years of ground-based remote sensing measurements at the ARM NSA Barrow site, Zhao (2011) shows that Arctic mixed-phase clouds in springtime have larger IWPs and smaller supercooled liquid water fraction than the other three seasons, which might be related to there being more dust events observed with lidar depolarization measurements during springtime that provide effective INPs for ice nucleation in clouds. The significant seasonal variations of ice production and their correspondence with dust occurrence in northern mid- and high-latitude mixed-phase clouds suggest that aerosol-dependent ice concentration parameterizations need to be used in GCMs and improved aerosol (especially dust) simulations are required in order to improve global mixed-phase cloud simulations, especially over the Northern Hemisphere.

## 4. Summary

Four years of collocated CALIPSO lidar and CloudSat radar measurements together with other ancillary A-Train products during 2006-2010 are analyzed to study primary ice particle production characteristics in single-layer mid-level stratiform mixed-phase clouds on a global scale. Mid-level stratiform mixed-phase clouds have a simple dynamic environment and straightforward ice growth trajectory that enables using $Z_e$ measurements to quantitatively infer ice number

concentration variations. We carefully isolate factors that impact ice diffusional growth and measured cloud layer radar $Z_e$ by focusing on mid-level stratiform mixed-phase clouds with same CTT and similar LWPs. We also analyzed atmospheric pressure impacts. Together with MODIS LWP retrievals and an improved thin dust layer detection algorithm, we connect the observed ZL differences and ice concentration variations to aerosol (especially dust) activities on a global scale.

5      Using the large dataset, we show that for similar clouds in terms of CTT and LWP, northern hemisphere latitude bands have ZL that are ~ 1 to 8 dBZ larger than their counterparts in southern hemisphere for a given CTT. After removing contributions from atmospheric pressure differences, ZL is still 6 dBZ larger that suggests a factor of 4 higher ice number concentrations (on average) over northern mid- and high latitudes than their southern counterparts. The systematically larger ZL and higher ice number concentrations in mid-level stratiform mixed-phase clouds over the Northern Hemisphere are
10 possibly related to larger background aerosol loadings. LWP has a significant impact on measured ZL, but we show that within a given narrow LWP range, mean ZL over northern latitude bands is always larger than their southern counterparts. Furthermore, we show that the northern mid- and high latitudes have dramatic seasonal variations in ZL, where ZL can be up to 6 dBZ larger in springtime than in wintertime. This might be related to more dust events during springtime that provide effective INPs for ice nucleation in clouds. Since mixed-phase cloud property evolution is strongly dependent on ice number
15 concentration, our study suggests that aerosol-dependent ice concentration parameterizations are required in GCMs in order to improve global mixed-phase cloud simulations. The results in this study can be used to evaluate global ice concentrations in mixed-phased clouds and aerosol impacts simulated by GCMs.

**Acknowledgements**

This study was funded by the U.S. Department of Energy (DOE) under grant DE-SC0012704 and the NASA under grant NNX13AQ41G. We thank the CloudSat data group at the CloudSat Data Processing Center and CALIPSO data group at the NASA Langley Atmospheric Sciences Data Center. The CloudSat and CALIPSO data used in this study can be downloaded from, respectively, http://www.cloudsat.cira.colostate.edu and https://eosweb.larc.nasa.gov/order-data.

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

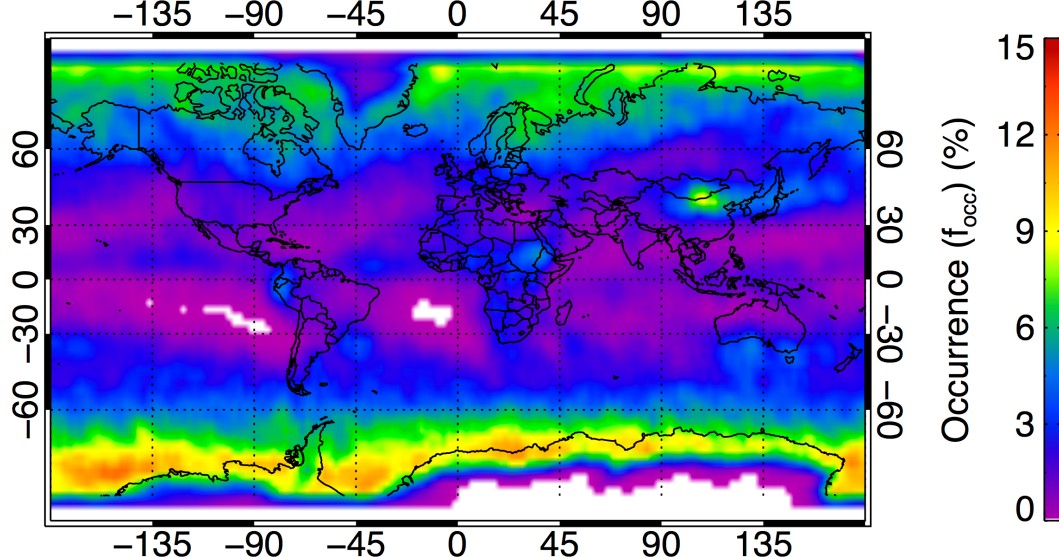

**Figure 1.** Global distribution of single-layer mid-level stratiform cloud occurrence frequency from four-years of collocated CALIOP and CPR measurements.

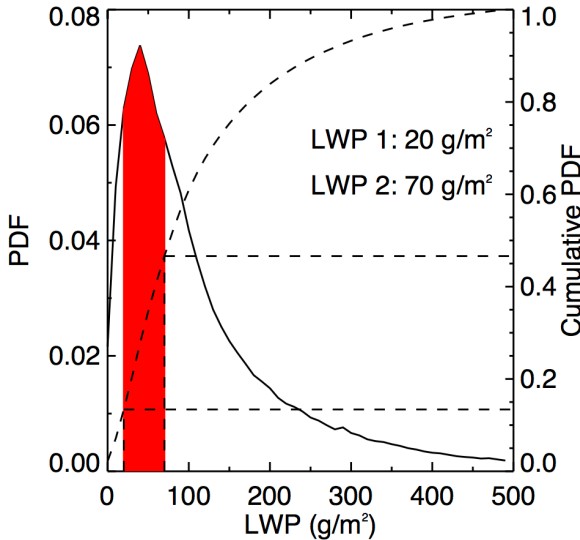

**Figure 2.** The probability distribution function (PDF) of LWP for single-layer mid-level stratiform clouds from MODIS retrievals. The red area indicates the third of the cumulative distribution that is centered on the peak of the LWP PDF. Given in the figure is LWP 1, the value for the lower third, and LWP 2, the value at the upper third.

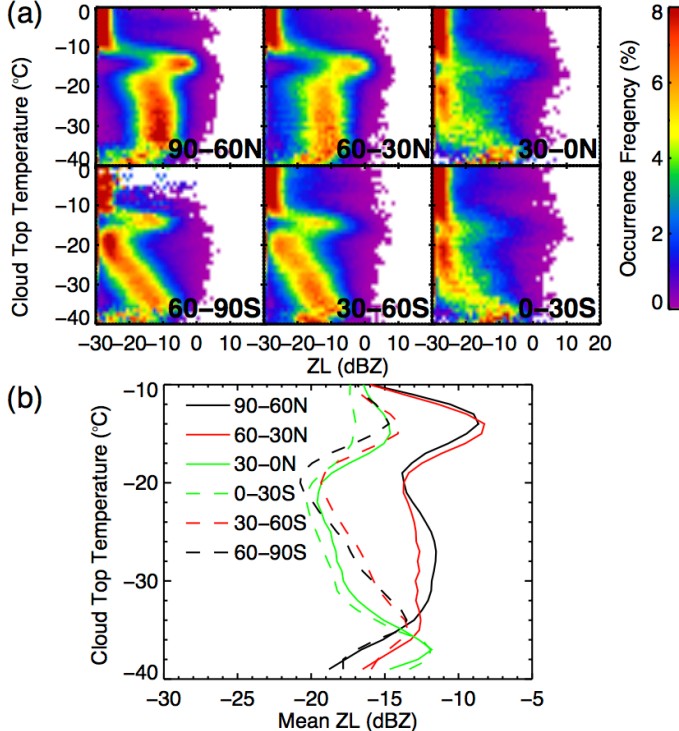

**Figure 3**. Global, annual-average, mid-level mixed-phase stratiform cloud ice production statistics. Results for six, 30°-latitude bands are shown covering the northern and southern tropical, mid-, and high-latitude regions. Cases are restricted so that the supercooled liquid water path is within the range between 20 g/m² and 70 g/m². (a) Cloud distributions are in terms of cloud top temperature (CTT) and layer-maximum radar reflectivity (ZL), a proxy for ice production. (b) Mean ZL of clouds as a function of CTT. Distributions are normalized at each CTT bin. The bin sizes for CTT and ZL are 1°C and 1 dBZ, respectively.

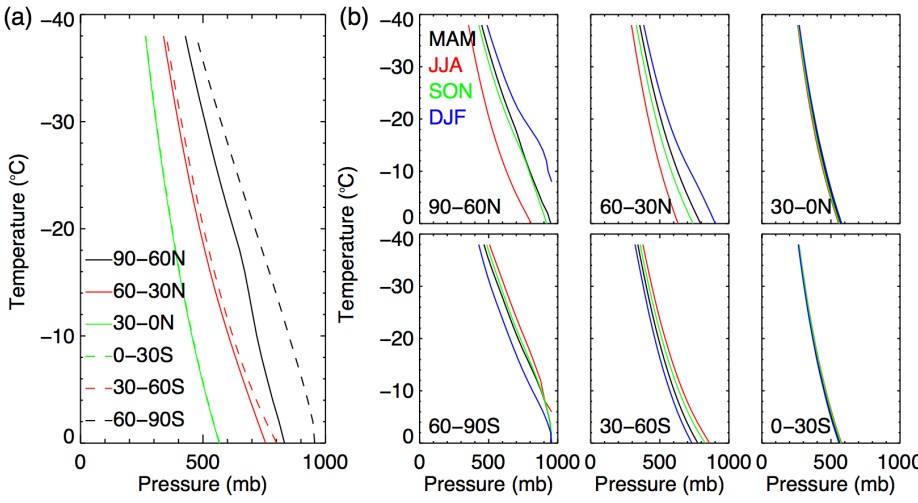

**Figure 4.** Atmospheric pressure profiles with subfreezing temperatures for (a) the six latitude bands and (b) their seasonal variations. MAM stands for March-April-May, JJA for June-July-August, SON for September-October-November, DJF for December-January-February.

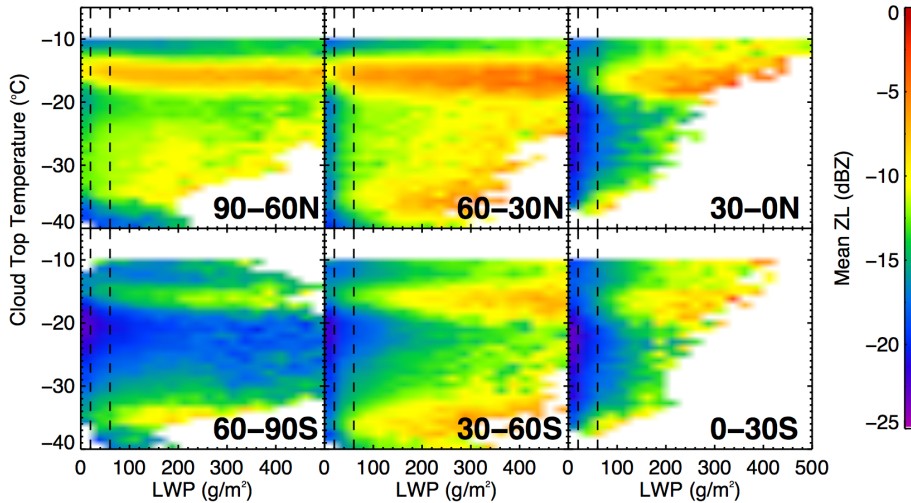

**Figure 5.** Mean of the layer-maximum radar reflectivity (ZL) of mid-level stratiform mixed-phase clouds as a function of cloud top temperature and liquid layer path (LWP). Results shown for six latitude bands as in Fig. 1. The dashed lines are the narrow range of LWP between 20-70 g/m$^2$.

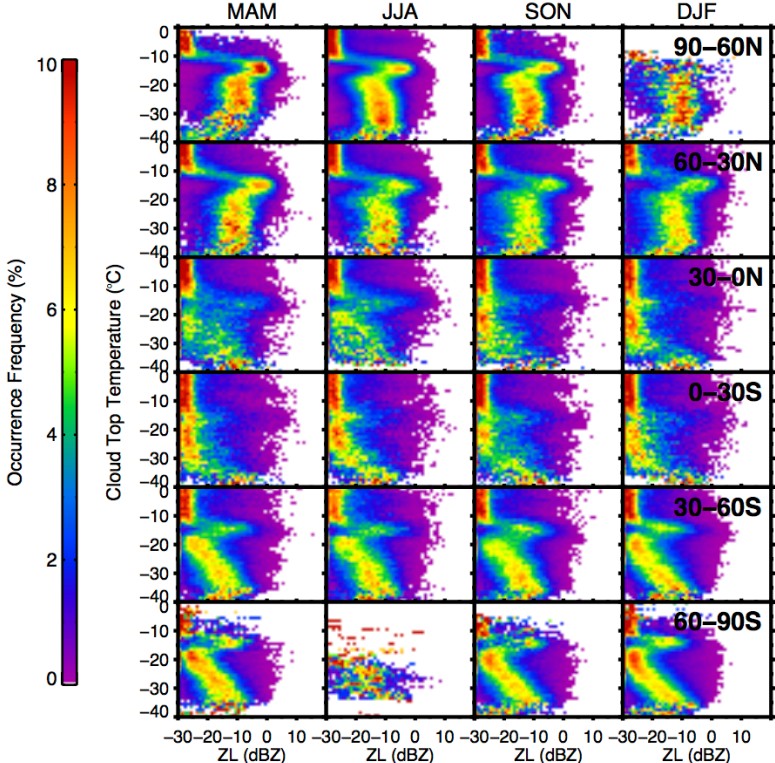

**Figure 6**. Similar to Figure 3, except for the seasonal variation in stratiform mixed-phase cloud ice production statistics.

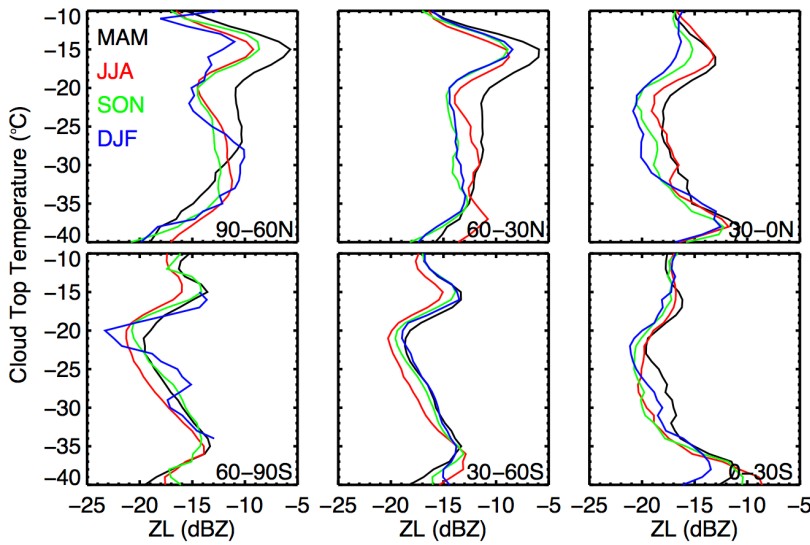

**Figure 7**. Similar to Figure 3, except for the seasonal variations in mean ZL of clouds as a function of CTT.

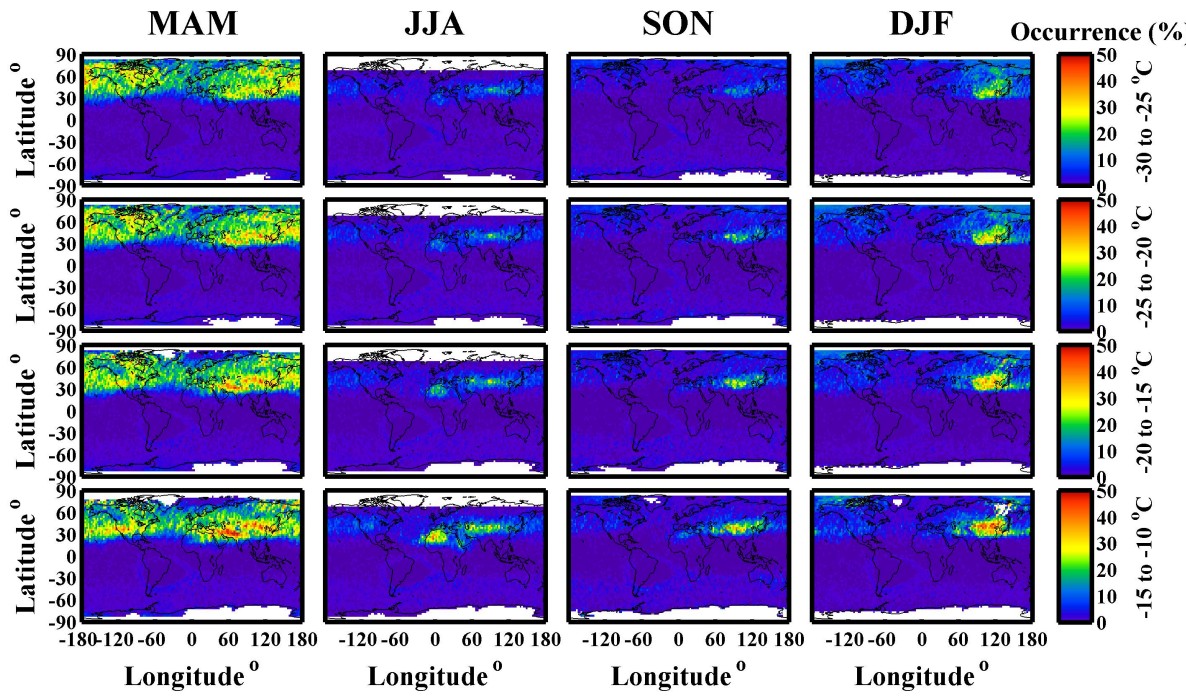

**Figure 8.** Distributions of global dust occurrences and their seasonal variations at different sub-freezing temperature ranges based on the dust dataset developed by Luo et al. (2014). Temperature ranges are given at the right. Each column is for a season, with the abbreviations described in Figure 4.

