# Peer review of "Ice Particle Production in Mid-level Stratiform Mixed-phase Clouds Observed with Collocated A-Train Measurements"

_Atmospheric Chemistry and Physics, 2017_

## Referee Comment (RC1) · P. Seifert (Referee) · 17 Nov 2017

The underlying manuscript of Zhang et al. is a follow-up publication in a series from the first author using combined CALIPSO and Cloudsat observations to investigate the structure of stratiform mixed-phase clouds on a global scale. The approach has been extended with respect to the previous studies by taking MODIS-retrieved LWP as additional constraint of microphysical properties of the observed clouds.

The presented results are of substantial value for the atmospheric science community, given that the study suggests the presence of a considerable hemispheric contrast in the ice formation efficiency in stratiform mixed-phase clouds. In principle, the dataset

seems to be well characterized and the data analysis methods can be considered mature, considering that several related studies were published by Zhang et al. since 2010.

Consequently, there are only a few critics points with respect to the technical implementation of the study. Nevertheless, the implied relevance of the results and the conclusions drawn are way beyond the data basis provided by the authors. This is a major issue. Basically, the message of the dataset is clear –> There is much less ice observed in stratiform mixed-phase clouds in the southern hemisphere. However, the conclusions are too linearly pointing toward aerosol effects. More efforts should be put by the authors on either supporting their strong conclusion, or, on providing a more brought discussion that includes other effects besides the aerosols. More details will follow below in the itemized review comments.

Major comments:

Technically:

1) What is the equation for the relationship between N_ice and Z? It should be presented because this relationship is discussed quite often. Which other parameters go into this equation? What is their role in the determination of Z? E.g., particle size. How much would particle size need to vary in order to explain the observed reflectivity differences? This is likely only a few percent due to the D^6 relationship. May there be any ice growth processes that could explain such a slight hemispheric difference in the crystal size? What if the cloud height and thus the pressure level of cloud formation varies regionally and seasonally? See, for example, Chapter 13.3. of the book of Pruppacher&Klett, 1997 (Fig. 13-29) that presents that the diffusional growth rate of ice crystals varies by up to 100% between pressures of 1000 mb and 500mb.

2) Why is layer maximum ZL used as reference value? Why not mean or, as done by Bühl et al., 2016, the value closest to the liquid layer?

3) Figure 5: When doing hemispheric studies it is not straightforward to use seasons. Better is to use month ranges and then refer to boreal and austral seasons in the text.

4) P 3, L 14: "usually decoupled" is a very vague statement. How often are clouds coupled to the surface or to the planetary boundary layer? This could be easily checked by using global model datasets such as GDAS1 which also provide an estimate of the mixing layer height. It would be a good test to investigate surface effects by excluding/including cloud layers touching the atmosphere-ground mixing layer from/into the statistics. Is there a hemispheric/seasonal variability of potential surface effects?

5) P3, L24-25: Sassen et al. 2012 showed strong effects of specular reflection on the CALIPSO measurements before it was tilted to $3°$ off-zenith. The authors argue on P4, L21ff that this does not affect the lidar-based liquid cloud determination. Was there an actual check performed to evaluate this assumption? The signal of CALIOP is known to attenuate quickly, also under compact cirrus conditions. Figure 3 in Sassen et al, 2012 shows a dramatic change in the relationship between LDR and temperature between nadir and off-nadir pointing, especially at T>-30°C.

Comments regarding the argumentation:

1) How do the different data analysis methods of Zhang 2010, 2012 and the current one differ? In the current version it is only argued that the current study differs from the 2010 study by considering only single-layer clouds. But can this explain why the results are so different? I would be happy to see some more text dealing with the cross-evaluation of the different studies. Perhaps a table would help to clarify methodological differences.

2) It should be noted that strong hemispheric differences in het. ice formation efficiency were already presented by Kanitz et al., 2011. Since the way of data analysis is similar to the presented study it would be worth mentioning it.

3) P7, L4-13: There are a lot of statements given in this paragraph. "crystals reside

longer", "grow larger by the WBF process", "larger ice growth rate by accretion". Are there references available supporting these statements?

4) I personally strongly support the conclusion given on P8, L16ff. However, are there really no other effects beside aerosol properties that should be discussed? I strongly recommend to at least point to the possibility that the difference in the reflectivities can be either attributed to large changes in the number OR to very small changes in the size of the ice crystals. The evolution of ice crystals depends on a multitude of constraints. . .just take a look into Pruppacher&Klett, 1997 or into modeling studies of mixed-phase clouds, such as the ones of Ann Fridlind or Morrisson et a. 2012. Also the studies of Korolev and/or Field show that cloud dynamics can have a strong effect on the evolution of the ice crystals. Constraining LWP is already a great leap forward, but other environmental parameters such as cloud pressure or the relationship between the clouds and the planetary boundary layer are just a few examples of possible additional factors. Consider also, as another example: Average CCN concentrations in the atmosphere over the Southern-hemispheric (SH) Oceans are only one fifth of the northern-hemispheric values (Yum et al. 2004). Assuming constant cloud depth and liquid water path, much fewer but much larger droplets can be expected in the SH clouds. Heterogeneous freezing parameterizations (especially immersion freezing) rely mainly on temperature and aerosol properties but not on droplet size (See, e.g., Demott et al., 2010). Thus, having much less droplets available for ice nucleation will result in correspondingly lower ice crystal concentrations, even in the absence of any aerosol effect. This pathway could also contribute to the apparently less efficient ice formation over the SH. There are still a lot of unknowns that need to be resolved before we can actually pin down the observations solely to aerosol effects. I'm looking forward to a lot of future studies dedicated to this key question of current atmospheric research.

Minor comments:

P2, L 16: An impressive demonstration of the lifetime effect as a function of temperature is also given by Bühl et al., 2016.

[Figure]

References:

Bühl, J., Seifert, P., Myagkov, A., and Ansmann, A.: Measuring ice- and liquid-water properties in mixed-phase cloud layers at the Leipzig Cloudnet station, Atmos. Chem. Phys., 16, 10609-10620, https://doi.org/10.5194/acp-16-10609-2016, 2016.

DeMott, P.J, A. J. Prenni, X. Liu, S. M. Kreidenweis, M. D. Petters, C. H. Twohy, M. S. Richardson, T. Eidhammer, and D. C. Rogers Predicting global atmospheric ice nuclei distributions and their impacts on climate, PNAS 2010 107 (25) 11217-11222; published ahead of print June 7, 2010, doi:10.1073/pnas.0910818107

Kanitz, T., P. Seifert, A. Ansmann, R. Engelmann, D. Althausen, C. Casiccia, and E. G. Rohwer (2011), Contrasting the impact of aerosols at northern and southern midlatitudes on heterogeneous ice formation, Geophys. Res. Lett., 38, L17802, doi:10.1029/2011GL048532.

Korolev, A. and P.R. Field, 2008: The Effect of Dynamics on Mixed-Phase Clouds: Theoretical Considerations. J. Atmos. Sci., 65, 66–86, https://doi.org/10.1175/2007JAS2355.1

Pruppacher, Hans R & Klett, James D., 1940- (1997). Microphysics of clouds and precipitation (2nd rev. and enl. ed). Kluwer Academic Publishers, Dordrecht ; Boston

Sassen, K., V. K. Kayetha, and J. Zhu (2012), Ice cloud depolarization for nadir and off-nadir CALIPSO measurements, Geophys. Res. Lett., 39, L20805, doi:10.1029/2012GL053116.

Yum, S. S., and J. G. Hudson (2004), Wintertime/summertime contrasts of cloud condensation nuclei and cloud microphysics over the Southern Ocean, J. Geophys. Res., 109, D06204, doi:10.1029/2003JD003864.

---

## Referee Comment (RC2) · Anonymous Referee #2 · 30 Nov 2017

In their paper, Zhang et al. analyse collocated CloudSat and CALIPSO lidar measurements between 2006 and 2010 to study ice number concentration in stratiform mixed-phase clouds. They divide the global data set into six latitude bands (northern and southern tropical, mid- and high-latitudes) for their analysis. In general the paper is well written and the results are of interest to the community. However, the method needs further explanation and the analysis/interpretation should take into account differences in the macro- and microphysical properties of Arctic and mid-latitude clouds. The paper needs major revision before it can be published in ACP. Page and line number below refer to the document uploaded by the authors.

[Figure]

Comments:

In the introduction the authors should discuss the specific characteristic of Arctic mixed-phase clouds (e.g. observation, life time and limited CCN). For example the observed CCN concentrations in Arctic mixed phase clouds are usually of the order of 10 cm$-3$ (rarely as high as 100 cm$-3$) and sometimes less than 1 cm$-3$ (Birch et al. 2012). This is in contrast to lower latitudes where typical concentrations range from approximately 100 cm$-3$ to several 1000 cm$-3$ in the marine environment (Raes et al., 2000). Such low CCN number concentrations affect cloud droplet size spectra, and hence, radiative properties of these clouds will differ from those at mid- latitudes. Mauritsen et al. (2011) argue that cloud formation is frequently limited by CCN availability in the central Arctic. They use the term "tenuous cloud regime" to describe situations in between an abundance of aerosol needed to form clouds and a hypothetical situation where aerosols are absent and cloud formation does only occur at very high super-saturation (âĹij400% relative humidity). Further, Arctic mixed-phase clouds are governed by a combination of local and large-scale processes (Morrison et al., 2012). At the small scale, the Wegener–Bergeron– Findeisen (WBF) process is one of the main mechanisms responsible for ice crystal growth at the expense of super-cooled water droplets (Bergeron, 1935; Findeisen, 1938; Wegener, 1911). Such a mechanism leads to a rapid glaciation of mixed-phase clouds. On the other hand, dynamical processes, such as turbulence or entrainment may facilitate the formation of new super-cooled water droplets. For example, resupply of water vapour from the surface or from entrainment of moisture from above the clouds may contribute to the continuous formation of liquid droplets. The coupling of various processes is, thus, necessary to maintain the unstable equilibrium between liquid droplets and ice crystals within Arctic mixed-phase clouds (Mioche et al 2015). This may explain the long lifetime of Arctic mixed-phase clouds, which can last up to several days or weeks. (Shupe, 2011; Verlinde et al., 2007; Morrison et al., 2012). Previous studies of Korolev et al. (2003), Korolev and Isaac, (2003) and Korolev (2007) also point out that the lifetime of Arctic mixed-phase depends on local thermodynamical conditions or is linked to cloud dynamics. Local

and long-range dynamic processes (aerosol, heat and moisture transport) also have a significant impact on Arctic mixed-phase cloud formation and properties (Cesana et al. 2012, Morrison et al. 2012).

Page 5, line 129-133. Why do you use max Ze and not mean Ze as Zhang et al. (2014)? Your results in Figure 3b (90-60 N) are similar to Figure 10 in Zhang et al. (2014) but shifted to larger values. What is the effect of using max or mean on the relationship between Ze and ice number concentration? Is Ze normally distributed to allow for a retrieval of a relationship to aerosol concentrations and the subsequent statistical analysis?

Page 6, line 156 to 160: Does the selected LWP range (20 to 70 gm-2) have an effect on the statistics for high-latitude clouds? Arctic mixed-phase clouds usually peak at lower LWP (Tjernström et al., 2012).

Page 6 six latitudes bands: Can you please repeat your analysis for an Arctic latitude band (> 70°). Figure 1 shows the highest occurrence of mixed-phase clouds over the ocean in the Arctic and Antarctic (> 70° and <-70°) while Figure 7 shows aerosol occurrence beyond 70° that is lower than at latitudes below 70° in spring. Your results might be biased by your choice of latitude band, i.e. the results for the latitude bands 60-90° might be dominated by the signals from between 60 and 70°. In other words: I don't think that there is so much dust in the Arctic that it could have such a strong effect in the clouds (see comments regarding the Introduction).

Page7, line 221-222, supercooled cloud fraction, [] lowest during springtime: Is the occurrence frequency of mixed-phase clouds according to Figure 5 not the highest in spring at high-latitudes?

Page 7, line 187, delete greater

References:

Birch, C.E. et al., 2012. Modelling atmospheric structure, cloudand their response to

CCN in the central Arctic: ASCOS case studies. Atmospheric Chemistry and Physics, 12(7), pp.3419–3435.

Cesana, G., Kay, J. E., Chepfer, H., English, J. M. and de Boer, G.: Ubiquitous low-level liquid-containing Arctic clouds: New observations and climate model constraints from CALIPSO-GOCCP, Geophys. Res. Lett., 39, L20804, 1–6, doi:10.1029/2012GL053385, 2012.

Korolev, A. V., G. A. Isaac, S. G. Cober, J. W. Strapp, and J. Hallett, 2003: Microphysical characterization of mixed- phase clouds. Quart. J. Roy. Meteor. Soc., 129, 39–55.

Korolev, A. V., and G. Isaac, 2003: Phase transformation of mixed- phase clouds.Quart. J. Roy.Meteor.Soc., 129, 19–38,doi:10.1256/ qj.01.203.

Mauritsen, T. & Sedlar, J., 2011. An Arctic CCN-limited cloud-aerosol regime. Atmospheric . . ., 11(1), pp.165–173.

Mioche, G. et al., 2015. Variability of mixed-phase clouds in the Arctic with a focus on the Svalbard region: A study based on spaceborne active remote sensing. Atmospheric Chemistry and Physics, 15, pp.2445–2461.

Morrison, H., G. de Boer, G. Feingold, J. Harrington, M. D. Shupe, and K. Sulia, 2012: Resilience of persis- tent Arctic mixed-phase clouds. Nat. Geosci., 5, 11–17

Raes, F., Van Dingenen, R., Vignati, E.,Wilson, J., Putaud, J. P., Se- infeld, J. H., and Adams, P.: Formation and cycling of aerosols in the global troposphere, Atmos. Environ., 34, 4215–4240, 2000.

Shupe, M.D., 2011. Clouds at Arctic Atmospheric Observatories. Part II: Thermodynamic Phase Characteristics. Journal of Applied Meteorology and Climatology, 50(3), pp.645–661. Verlinde, J., and Coauthors, 2007: The Mixed-Phase Arctic Cloud Experiment (M-PACE). Bull. Amer. Meteor. Soc., 88, 205–220.

Tjernström, M. et al., 2012. Meteorological conditions in the central Arctic summer

during the Arctic Summer Cloud Ocean Study (ASCOS). Atmospheric Chemistry and Physics, 12(15), pp.6863–6889.

Zhang, D. et al., 2014. Ice Concentration Retrieval in Stratiform Mixed-Phase Clouds Using Cloud Radar Reflectivity Measurements and 1D Ice Growth Model Simulations. Journal of the Atmospheric Sciences, 71, pp.3613–3635.

---

## Author Comment (AC1) · 25 Jan 2018

*P. Seifert (Referee)* seifert@tropos.de

*The underlying manuscript of Zhang et al. is a follow-up publication in a series from the first author using combined CALIPSO and Cloudsat observations to investigate the structure of stratiform mixed-phase clouds on a global scale. The approach has been extended with respect to the previous studies by taking MODIS-retrieved LWP as additional constraint of microphysical properties of the observed clouds.*

*The presented results are of substantial value for the atmospheric science community, given that the study suggests the presence of a considerable hemispheric contrast in the ice formation efficiency in stratiform mixed-phase clouds. In principle, the dataset seems to be well characterized and the data analysis methods can be considered mature, considering that several related studies were published by Zhang et al. since 2010.*

*Consequently, there are only a few critics points with respect to the technical implementation of the study. Nevertheless, the implied relevance of the results and the conclusions drawn are way beyond the data basis provided by the authors. This is a major issue. Basically, the message of the dataset is clear –> There is much less ice observed in stratiform mixed-phase clouds in the southern hemisphere. However, the conclusions are too linearly pointing toward aerosol effects. More efforts should be put by the authors on either supporting their strong conclusion, or, on providing a more brought discussion that includes other effects besides the aerosols. More details will follow below in the itemized review comments.*

**Author Response: We thank the reviewer for the very helpful comments; we carefully revised the manuscript according to the reviewer's comments as presented below.**

*Major comments:*

*Technically:*

*1) What is the equation for the relationship between N_ice and Z? It should be presented because this relationship is discussed quite often. Which other parameters go into this equation? What is their role in the determination of Z? E.g., particle size. How much would particle size need to vary in order to explain the observed reflectivity differences? This is likely only a few percent due to the D^6 relationship. May there be any ice growth processes that could explain*

*such a slight hemispheric difference in the crystal size? What if the cloud height and thus the pressure level of cloud formation varies regionally and seasonally? See, for example, Chapter 13.3. of the book of Pruppacher&Klett, 1997 (Fig. 13-29) that presents that the diffusional growth rate of ice crystals varies by up to 100% between pressures of 1000 mb and 500mb.*

**Author Response: We thank the reviewer for the very constructive comments. It is quite challenging to retrieve ice number concentration ($N_{ice}$) from radar reflectivity ($Z_e$) measurements. Zhang et al. (2014) developed a method to estimate $N_{ice}$ in stratiform mixed-phase clouds by using combined $Z_e$ measurements and 1-D ice-growth model simulations. Cloud top temperature, liquid water path (LWP), and vertical air motion are required as inputs in their algorithms and sensitivity tests show that they all have important impacts on $N_{ice}$ estimations. Due to large uncertainties in the MODIS-derived LWP for mixed-phase clouds (Adhikari and Wang 2013), $N_{ice}$ estimations in mixed-phase clouds using A-Train satellite measurements are not available at this stage. We added this discussion to page 6 lines 15-20 in the text.**

**To account for the observed $Z_e$ differences of up to 8 dBZ in mid-level stratiform mixed-phase clouds and assuming $N_{ice}$ does not change, ice particle size needs to vary approximately 35% for a given cloud top temperature and similar narrow LWP range. We thank the reviewer for pointing out that atmospheric pressure might have systematic impacts on the ice diffusional growth. Takahashi et al. (1991) show that the mass growth rate at 860 mb is approximately 30% larger than at 1010 mb due to the variation of diffusivity of water vapor in air with the pressure difference. However, from Figure 20 in their paper, the mass growth difference due to pressure difference is much smaller than that due to temperature deference. Within the Rayleigh scattering regime, radar reflectivity is proportional to the square of ice crystal mass. Therefore, the 30% difference in mass causes approximately a 2 dBZ difference in $Z_e$. As suggested by the reviewer, we investigated the regional and seasonal differences of atmospheric pressure at given subfreezing temperatures using four years of the ECMWF-AUX product between 2006 and 2010.**

[Figure]

*Atmospheric pressure profiles with subfreezing temperatures for the six latitude bands (a) and their seasonal variations (b).*

From the above figure (a), hemispheric differences in atmospheric pressure profiles are negligible over mid- and low-latitude bands, and range from 40 mb to 140 mb over high latitudes. Therefore, pressure-level differences have negligible contribution to the hemispheric ZL differences over mid- and low latitude bands, and contribute less than 2 dBZ to the observed hemispheric ZL differences over high latitudes. Low-latitude bands have about 140 mb and 280 mb lower pressure than mid- and high-latitude bands, respectively, indicating the that ZL differences between low latitudes and mid- and high latitudes will be 1-2 dBZ larger after removing pressure difference effects. As shown in figure (b), northern mid- and high latitude atmospheric pressure profiles have 220 mb and 160 mb seasonal variations; while southern latitudes and northern low-latitudes have negligible seasonal variations. Overall, regional and seasonal variations of atmospheric pressure profiles cause less than 2 dBZ variations in ZL and have small impacts on our conclusions. We added the above figure in the revision (new Fig. 4) and added a discussion of the atmospheric pressure regional and seasonal differences and their impacts in the manuscript (pages 7-8).

References:

Adhikari, L., and Wang, Z.: An A-train satellite based stratiform mixed-phase cloud retrieval algorithm by combining active and passive sensor measurements, British Journal of Environment and Climate Change, 3(4), 587-611, 2013.

Zhang, D., Wang, Z., Heymsfield, A. J., Fan, J., and Luo, T.: Ice Concentration Retrieval in Stratiform Mixed-Phase Clouds Using Cloud Radar Reflectivity Measurements and 1D Ice Growth Model Simulations, J. Atmos. Sci., 71(10), 3613–3635, doi:10.1175/JAS-D-13-0354.1, 2014.

Takahashi, T., Endoh, T., Wakahama, G., and Fukuta, N.: Vapor diffusional growth of freefalling snow crystals between −3 and −23 ∘C, J. Meteorol. Soc. Jpn., 69, 15–30, 1991.

*2) Why is layer maximum ZL used as reference value? Why not mean or, as done by Bühl et al., 2016, the value closest to the liquid layer?*

**Author Response: In page 4 line 7, we pointed out that 'the CPR has an effective vertical resolution of about 480 m and is oversampled at 240 m vertical resolution'. Therefore, to remove possible $Z_e$ measurements above the top of liquid layer, we use the maximum $Z_e$ within 500 m below the CALIOP-detected liquid-dominated layer top as the reference value. In this revision, we emphasized the impacts of the coarse CPR resolution and oversampling technique by adding 'oversampled at 240 m from the effective vertical range resolution of 480 m' to page 5 line 24. Bühl et al., (2016) use the $Z_e$ value closest to the liquid layer base with ground-based high vertical resolution (30 m) radar measurements. However, this is difficult with A-Train satellite measurements as the CPR has a coarse vertical resolution and the liquid layer at the top quickly attenuates CALIOP signals, preventing reliable detection of the liquid layer base. We also added this discussion to page 5 lines 19-22 in the text.**

*3) Figure 5: When doing hemispheric studies it is not straightforward to use seasons. Better is to use month ranges and then refer to boreal and austral seasons in the text.*

**Author Response: We revised the figure as suggested and refer them to boreal and austral seasons in the text.**

*4) P 3, L 14: "usually decoupled" is a very vague statement. How often are clouds coupled to the surface or to the planetary boundary layer? This could be easily checked by using global model datasets such as GDAS1 which also provide an estimate of the mixing layer height. It would be a good test to investigate surface effects by excluding/including cloud layers touching the atmosphere-ground mixing layer from/into the statistics. Is there a hemispheric/seasonal variability of potential surface effects?*

**Author Response: We thank the reviewer for the suggestions. Ice processes in atmospheric clouds are very complicated. In this study, we aim to analyze primary heterogeneous ice formation characteristics in supercooled clouds with satellite remote sensing measurements by focusing on mid-level stratiform clouds. Turbulent vertical mixing within the boundary layer has significant impacts on ice particle growth and the lifecycle of mixed-phase clouds, making it very challenging to study primary ice formation with radar $Z_e$ measurements. Therefore, we did not include boundary layer mixed-phase clouds in this study. To make it more clear, we deleted "usually" and added "Mid-level supercooled Cloud" to emphasize the targeted clouds page 3 line 24 in this revision.**

**Investigating surface effects on mixed-phase clouds is important but is out of the scope of this paper. Using ground-based remote sensing, including Doppler lidar and radar spectral measurements, turbulent vertical mixing impacts on polar boundary layer mixed-phase cloud ice particle growth, secondary ice generation, supercooled liquid water fraction, and cloud lifecycle are under investigation in our group.**

*5) P3, L24-25: Sassen et al. 2012 showed strong effects of specular reflection on the CALIPSO measurements before it was tilted to 3◦ off-zenith. The authors argue on P4, L21ff that this does not affect the lidar-based liquid cloud determination. Was there an actual check performed to evaluate this assumption? The signal of CALIOP is known to attenuate quickly, also under compact cirrus conditions. Figure 3 in Sassen et al, 2012 shows a dramatic change in the relationship between LDR and temperature between nadir and off-nadir pointing, especially at T>-30◦C.*

**Author Response: We agree with the reviewer that horizontally oriented ice crystals present a challenge for lidar-based liquid cloud determination because they also cause large lidar backscattering and small depolarization ratio measurements. However, ground-based lidar measurements show that horizontally oriented ice crystals have a weak attenuation of lidar signals and lidar signals can penetrate ice cloud layer up to a few kilometers (Hogan et al., 2003; Balin et al., 2011). This provides an alternative way to effectively detect liquid layer using lidar signal profile. Wang (2013, Figure 10) shows that this approach correctly determines liquid clouds in terms of layer mean depolarization ratio and integrated backscattering coefficient. In addition, collocated MODIS cloud LWP greater than 10 g/m$^2$ is used to guarantee the detection of a liquid-dominated layer. We added this statement in page 5 lines 8-11 in the text.**

**References:**

**Balin, Y. S., Kaul, B. V., Kokhanenko, G. P., & Penner, I. E. (2011). Observations of specular reflective particles and layers in crystal clouds. *Optics express*, *19*(7), 6209-6214.**

**Hogan, R. J., Illingworth, A. J., O'connor, E. J. and Baptista, J. P. V. P. (2003), Characteristics of mixed-phase clouds. II: A climatology from ground-based lidar. Q.J.R. Meteorol. Soc., 129: 2117–2134. doi:10.1256/qj.01.209.**

**Wang, Z.: Level 2 combined radar and lidar cloud scenario classification product process description and interface control document. Jet Propulsion Laboratory Tech. Rep. D-xxxx, 61 pp.**

*Comments regarding the argumentation:*

*1) How do the different data analysis methods of Zhang 2010, 2012 and the current one differ? In the current version it is only argued that the current study differs from the 2010 study by considering only single-layer clouds. But can this explain why the results are so different? I would be happy to see some more text dealing with the cross-evaluation of the different studies. Perhaps a table would help to clarify methodological differences.*

**Author Response: The reviewer is right that this research is a follow-up study of Zhang et al. (2010) and Zhang et al. (2012). The data analysis methods in this study are similar to that in Zhang et al. (2010) and Zhang et al. (2012). Zhang et al. (2010) present for the first time the climatology of mid-level stratiform clouds and their macrophysical properties using A-Train satellite measurements. Considering that mid-level stratiform clouds are ideal targets for studying primary ice formation and aerosol impacts, Zhang et al. (2012) quantitatively estimated dust impacts on ice production in mixed-phase clouds over the**

**'dust belt'**, a region including the North Africa, the Arabian Peninsula and East Asia. Taking advantage of improved understanding of ice diffusional growth in mixed-phase clouds as shown in Zhang et al. (2014) and improved thin dust layer detection with CALIOP by Luo et al. (2014), this study further uses A-Train satellite measurements to provide a global statistical analysis of ice production in mid-level stratiform mixed-phase clouds. We introduced Zhang et al. (2010) study in line 33 and Zhang et al. (2012) study in line 20 on page 3.

*2) It should be noted that strong hemispheric differences in het. ice formation efficiency were already presented by Kanitz et al., 2011. Since the way of data analysis is similar to the presented study it would be worth mentioning it.*

**Author Response: We thank the reviewer for the very helpful comment. We added discussions of Kanitz et al. (2011) study on page 8 lines 5-8.**

*3) P7, L4-13: There are a lot of statements given in this paragraph. "crystals reside longer", "grow larger by the WBF process", "larger ice growth rate by accretion". Are there references available supporting these statements?*

**Author Response: Direct observation of ice diffusional growth in clouds is difficult to obtain. Zhang et al., developed 1-D ice growth model to simulate ice diffusional growth in clouds and investigate LWP impacts. These statements are based on our fundamental understanding of the ice diffusional growth process and the 1-D ice growth model simulations. In this revision, we added the reference to Zhang et al. (2014) in the text.**

**Reference:**

**Zhang, D., Wang, Z., Heymsfield, A. J., Fan, J., and Luo, T.: Ice Concentration Retrieval in Stratiform Mixed-Phase Clouds Using Cloud Radar Reflectivity Measurements and 1D Ice Growth Model Simulations, J. Atmos. Sci., 71(10), 3613–3635, doi:10.1175/JAS-D-13-0354.1, 2014.**

*4) I personally strongly support the conclusion given on P8, L16ff. However, are there really no other effects beside aerosol properties that should be discussed? I strongly recommend to at least point to the possibility that the difference in the reflectivities can be either attributed to large changes in the number OR to very small changes in the size of the ice crystals. The evolution of ice crystals depends on a multitude of constraints. . .just take a look into Pruppacher&Klett, 1997 or into modeling studies of mixed-phase clouds, such as the ones of Ann Fridlind or Morrisson et a. 2012. Also the studies of Korolev and/or Field show that cloud dynamics can have a strong effect on the evolution of the ice crystals. Constraining LWP is already a great leap forward, but other environmental parameters such as cloud pressure or the relationship between the clouds and the planetary boundary layer are just a few examples of possible additional factors. Consider also, as another example: Average CCN concentrations in the atmosphere over the Southern-hemispheric (SH) Oceans are only one fifth of the northern-hemispheric values (Yum et al. 2004). Assuming constant cloud depth and liquid water path, much fewer but much larger droplets can be expected in the SH clouds. Heterogeneous freezing parameterizations (especially immersion freezing) rely mainly on temperature and aerosol*

*properties but not on droplet size (See, e.g., Demott et al., 2010). Thus, having much less droplets available for ice nucleation will result in correspondingly lower ice crystal concentrations, even in the absence of any aerosol effect. This pathway could also contribute to the apparently less efficient ice formation over the SH. There are still a lot of unknowns that need to be resolved before we can actually pin down the observations solely to aerosol effects. I'm looking forward to a lot of future studies dedicated to this key question of current atmospheric research.*

**Author Response: We thank the reviewer for the very constructive comments. We pointed out that "differences in $Z_e$ can be either attributed to large changes in ice number concentration or small changes in ice crystal size" in page 6 lines 8-10. In this study we carefully isolate factors that impact ice diffusional growth and particle sizes by focusing on mid-level stratiform mixed-phase clouds with same CTT and similar LWPs. We also analyzed atmospheric pressure impacts as suggested in the Technical comment #1. The motivation for targeting mid-level stratiform mixed-phase clouds include that they are not affected by strong turbulent vertical mixing within the boundary layer and have less complex dynamic environment and straightforward ice growth trajectory. We are aware that cloud dynamics have a strong effect on the evolution of the ice crystals, making it very challenging to study primary ice formation with radar $Z_e$ measurements. Therefore, we did not include boundary layer clouds in this study.**

**We agree with the reviewer that Southern-hemispheric Oceans have much lower CCN concentrations compared with low latitudes. However, the cloud droplet concentration is still several orders of magnitude higher than typical ice number concentrations and therefore the low CCN concentration condition is expected to have small impact on ice nucleation in cloud.**

**We agree with the reviewer that there are still a lot of unknowns related to ice processes in clouds and we toned down statements in the summary section.**

*Minor comments:*

*P2, L 16: An impressive demonstration of the lifetime effect as a function of temperature is also given by Bühl et al., 2016.*

**Author Response: We thank the reviewer for this helpful comment. We added discussion of the Bühl et al. (2016) study to page 2 lines 23-25 in the text.**

*Anonymous Referee #2*

*In their paper, Zhang et al. analyse collocated CloudSat and CALIPSO lidar measurements between 2006 and 2010 to study ice number concentration in stratiform mixed-phase clouds. They divide the global data set into six latitude bands (northern and southern tropical, mid- and high-latitudes) for their analysis. In general the paper is well written and the results are of interest to the community. However, the method needs further explanation and the analysis/interpretation should take into account differences in the macro- and microphysical properties of Arctic and mid-latitude clouds. The paper needs major revision before it can be published in ACP. Page and line number below refer to the document uploaded by the authors.*

**Author Response: We thank the reviewer for the helpful comments. We carefully revised the manuscript according to the reviewer's comments as presented below.**

*Comments:*

*In the introduction the authors should discuss the specific characteristic of Arctic mixed- phase clouds (e.g. observation, life time and limited CCN). For example the observed CCN concentrations in Arctic mixed phase clouds are usually of the order of 10 cm−3 (rarely as high as 100 cm−3) and sometimes less than 1 cm−3 (Birch et al. 2012). This is in contrast to lower latitudes where typical concentrations range from approximately 100 cm−3 to several 1000 cm−3 in the marine environment (Raes et al., 2000). Such low CCN number concentrations affect cloud droplet size spectra, and hence, radiative properties of these clouds will differ from those at mid- latitudes. Mauritsen et al. (2011) argue that cloud formation is frequently limited by CCN availability in the central Arctic. They use the term "tenuous cloud regime" to describe situations in between an abundance of aerosol needed to form clouds and a hypothetical situation where aerosols are absent and cloud formation does only occur at very high super-saturation (âˊLij400% relative humidity). Further, Arctic mixed-phase clouds are governed by a combination of local and large-scale processes (Morrison et al., 2012). At the small scale, the Wegener–Bergeron– Findeisen (WBF) process is one of the main mechanisms responsible for ice crystal growth at the expense of super-cooled water droplets (Bergeron, 1935; Findeisen, 1938; Wegener, 1911). Such a mechanism leads to a rapid glaciation of mixed-phase clouds. On the other hand, dynamical processes, such as turbulence or entrainment may facilitate the formation of new super-cooled water droplets. For example, resupply of water vapour from the surface or from entrainment of moisture from above the clouds may contribute to the continuous formation of liquid droplets. The coupling of various processes is, thus, necessary to maintain the unstable equilibrium between liquid droplets and ice crystals within Arctic mixed-phase clouds (Mioche et al 2015). This may explain the long lifetime of Arctic mixed-phase clouds, which can last up to several days or weeks. (Shupe, 2011; Verlinde et al., 2007; Morrison et al., 2012). Previous studies of Korolev et al. (2003), Korolev and Isaac, (2003) and Korolev (2007) also point out that the lifetime of Arctic mixed-phase depends on local thermodynamical conditions or is linked to cloud dynamics. Local and long-range dynamic processes (aerosol, heat and moisture transport) also have a significant impact on Arctic mixed-phase cloud formation and properties (Cesana et al. 2012, Morrison et al. 2012).*

**Author Response: We thank the reviewer for the very constructive comments. As suggested, we added a several sentences in the introduction part to discuss the specific**

**characteristics of polar mixed-phase clouds on page 2 lines 17-25.**

*Page 5, line 129-133. Why do you use max Ze and not mean Ze as Zhang et al. (2014)? Your results in Figure 3b (90-60 N) are similar to Figure 10 in Zhang et al. (2014) but shifted to larger values. What is the effect of using max or mean on the relationship between Ze and ice number concentration? Is Ze normally distributed to allow for a retrieval of a relationship to aerosol concentrations and the subsequent statistical analysis?*

**Author Response: We thank the reviewer for figuring out the differences. On page 4 line 7, we pointed out that 'the CPR has an effective vertical resolution of about 480 m and is oversampled at 240 m vertical resolution'. Therefore, to remove possible $Z_e$ measurements above the top of liquid layer, we use the maximum $Z_e$ within 500 m below the CALIOP-detected liquid-dominated layer top as the reference value. Zhang et al. (2014) use $Z_e$ profiles from ARM ground-based radar measurements, which have a much higher vertical resolution of 45 m. Therefore, layer-mean $Z_e$ is used in their study in order to be better compare with the 1-D ice growth model simulations. Another possible cause for mean $Z_e$ profile differences is that the CPR has a sensitivity of ~ -29 dBZ, while the ARM ground-based millimeter-wavelength radar has a higher sensitivity of ~ -60 dBZ, as shown in the figure below.**

[Figure]

*Stratiform mixed-phase cloud distributions in terms of CTT and $Z_e$ over the NSA Barrow site from: CloudSat CPR measurements (left) and ARM NSA KA-band cloud radar (KAZR) measurements (right). The distributions are normalized at each CTT.*

The reviewer is right that the distribution in Figure 3 is normalized at each cloud top temperature bin. We pointed this out in the caption of Figure 3.

*Page 6, line 156 to 160: Does the selected LWP range (20 to 70 gm-2) have an effect on the statistics for high-latitude clouds? Arctic mixed-phase clouds usually peak at lower LWP (Tjernström et al., 2012).*

**Author Response: We agree with the reviewer that Arctic mixed-phase clouds usually peak at low LWP compared with low latitudes. In this study, we chose the narrow LWP range centered on the LWP distribution peak to reduce the impacts of LWP variations on measured $Z_e$ and also to ensure enough samples are obtained. As long as the LWP range is same for all latitude bands at a given cloud temperature range, ice crystal growth in mid-level stratiform mixed-phase clouds are similar. As for getting enough samples, although the selected LWP range (20 to 70 g/m$^2$) is slightly off-peak of LWP distribution for Arctic mixed-phase clouds, we still have abundant samples because Arctic has high occurrence frequency of mixed-phase clouds. Therefore, the selected LWP range does not have an important impact on the statistics for high-latitude clouds.**

*Page 6 six latitudes bands: Can you please repeat your analysis for an Arctic latitude band (> 70◦). Figure 1 shows the highest occurrence of mixed-phase clouds over the ocean in the Arctic and Antarctic (> 70◦ and <-70◦) while Figure 7 shows aerosol occurrence beyond 70◦ that is lower than at latitudes below 70◦ in spring. Your results might be biased by your choice of latitude band, i.e. the results for the latitude bands 60-90◦ might be dominated by the signals from between 60 and 70◦. In other words: I don't think that there is so much dust in the Arctic that it could have such a strong effect in the clouds (see comments regarding the Introduction).*

**Author Response: We thank the reviewer for the helpful comments. We repeated analyses for Arctic latitude band (> 70$^o$) as shown below. Compared with Figure 7 in our manuscript, this higher Arctic latitude band (> 70$^o$) shows similar seasonal variations with maxima ZLs in spring, and also a similar magnitude of seasonal mean ZL differences.**

[Figure]

*Mid-level stratiform mixed-phase cloud distributions in terms of CTT and $Z_e$ for Arctic latitude band (>70$^0$) and their seasonal variations (left), and mean ZL seasonal variations (right). Winter daytime*

*measurements are sparse due to polar night and therefore its mean ZL is not plotted.*

**Indeed, elevated dust layers have approximately a 14% occurrence frequency during MAM as observed with ground-based lidar depolarization measurements at the ARM NSA Barrow site (latitude: 71.25 N), as shown in Figure 3.11 in Zhao (2011).**

**Reference:**

**Zhao, M.: The Arctic clouds from model simulations and long-term observations at Barrow, Alaska, PhD thesis, Univ. of Wyo., Laramie, 2011. https://search.proquest.com/docview/1354441520**

*Page7, line 221-222, supercooled cloud fraction, [] lowest during springtime: Is the occurrence frequency of mixed-phase clouds according to Figure 5 not the highest in spring at high-latitudes?*

**Author Response: The occurrence frequency of mixed-phase clouds relative to all liquid-containing supercooled clouds is the highest in spring at northern high-latitudes according to Figure 5 (now Figure 6). In Choi et al. (2010), supercooled cloud fraction (SCF) is derived as the ratio of liquid-containing supercooled clouds to all clouds at a given temperature range between -40 and 0 °C. Reductions in SCF indicate increases of glaciated cloud fractions. They therefore argue that strong correlations between mineral dust occurrence and reduction in SCF suggests that elevated mineral dust particles effectively glaciate supercooled clouds by providing abundant ice nucleation particles (INPs). Strictly, the definitions of SCF and occurrence frequency of mixed-phase clouds are not the same and should not be directly compared. We deleted "consistent with the results in Fig. 5 and 6 in our study" in the text.**

*Page 7, line 187, delete greater*

**Author Response: We deleted it as suggested.**

---

## Author Response (AR2)

Dear Editor:

Thank you very much for handling our manuscript. Point-by-point responses to each of the technical corrections are provided below. Our replies are highlighted in blue color while the original comments are black bold-faced.

Sincerely,

Damao Zhang

**Technical corrections:**

**- Please improve the resolution of your figures for publications. They are currently to coarse.**

We replotted all the figures with high resolution.

**- Please stick to the acronyms and signs once you introduced them. I am particularly referring to CALIPSO, CTT, IWP, and N_ice but there might be more. Please check the manuscript.**

We thank the editor for the comment. We removed duplicated acronym definitions (e.g. 'CALIPSO' in page 4, line1; 'IWPs' in page 9, line18). We stick to using CTT after its first definition in page 6, line 12 (except in figure captions so that readers can interpret the figures quickly by reading the captions). We also deleted the '$N_{ice}$' acronym definition and used 'ice number concentration' all the time.

**- Figure 6 is not really discussed and its message is transported by Figure 7. You might consider omitting Figure 6.**

We thank the editor for this comment. However, Fig. 6 is a key figure to show that observed larger ZL seasonal variations over northern latitude bands are statistically reliable. We point this out in the manuscript in page 8 line 29.

**- page 3, line 6: remove one of the stratiforms**

We removed one of the 'stratiforms'.

**- page 3, lines 8/9: use just 'INP', not 'INP aerosols'**

We changed it to 'INP' as suggested.

**- page 4, line 13: Winker et al. (2003) might not be the best reference for CALIOP**

We changed the reference to an updated one – Winker et al. (2007).

**- page 6, line 12: change 'are' to 'is'**

Corrected.

**- page 8, line 15: change 'are not able to' to 'cannot be'**

Corrected.